

# A new class of higher quantum Airy structures as modules of $\mathcal{W}(\mathfrak{gl}_r)$-algebras

**Vincent Bouchard[1⋆] and Kieran Mastel[2†]**

**1** Department of Mathematical & Statistical Sciences,
University of Alberta, 632 CAB Edmonton, Alberta, Canada T6G 2G1
**2** Department of Pure Mathematics, University of Waterloo,
200 University Avenue West, Waterloo, Ontario, Canada N2L 3G1

⋆ vincent.bouchard@ualberta.ca , † kmastel@uwaterloo.ca

## Abstract

Quantum $r$-Airy structures can be constructed as modules of $\mathcal{W}(\mathfrak{gl}_r)$-algebras via restriction of twisted modules for the underlying Heisenberg algebra. In this paper we classify all such higher quantum Airy structures that arise from modules twisted by automorphisms of the Cartan subalgebra consisting of products of disjoint cycles of the same length. An interesting feature of these higher quantum Airy structures is that the dilaton shifts must be chosen carefully to satisfy a matrix invertibility condition, with a natural choice being roots of unity. We explore how these higher quantum Airy structures may provide a definition of the Chekhov, Eynard, and Orantin topological recursion for reducible algebraic spectral curves. We also study under which conditions quantum $r$-Airy structures that come from modules twisted by arbitrary automorphisms can be extended to new quantum $(r+1)$-Airy structures by appending a trivial one-cycle to the twist without changing the dilaton shifts.



# 1    Introduction

The topological recursion of Chekhov, Eynard, and Orantin [10–12] is a formalism that can be used to solve various enumerative problems involving Riemann surfaces. It relies on the geometry of a spectral curve, which is realized as a branched covering of $\mathbb{P}^1$, and constructs generating functions for enumerative invariants via residue analysis on the spectral curve.

Kontsevich and Soibelman introduced the concept of quantum Airy structures in [1, 13] as an algebraic reformulation (and generalization) of the topological recursion. A quantum Airy structure is a set of second-order partial differential operators $H_i$ that satisfy certain specific conditions. The key result is that, under these conditions, there always exists a unique solution to the differential constraints $H_i Z = 0$, where $Z$ has the specific form of a generating function. It can be shown that the topological recursion of Chekhov, Eynard, and Orantin can be reformulated as a special case of quantum Airy structures [1, 5, 13].

Meanwhile, the original topological recursion of Chekhov, Eynard, and Orantin was also generalized in [7–9]. In the original formulation, the spectral curve was required to be a branched covering of $\mathbb{P}^1$ with only simple ramification points. The restriction was removed in [7–9], to allow for branched coverings with arbitrary ramifications. However, the resulting topological recursion is not a special case of quantum Airy structures anymore as originally formulated by Kontsevich and Soibelman.

This conundrum was resolved in [4], where the concept of higher quantum Airy structures was introduced. The main difference with the original quantum Airy structures of Kontsevich and Soibelman is that the restriction that the differential operators are second-order is relaxed, and differential operators of arbitrary order are considered. In particular, a large class of higher quantum Airy structures was constructed in [4] as modules for $\mathcal{W}(\mathfrak{g})$-algebras, where $\mathfrak{g}$ is a Lie algebra. It was then shown that the generalized topological recursion of [7–9] can be reformulated as a special case of higher quantum Airy structures realized as modules of $\mathcal{W}(\mathfrak{gl}_r)$-algebras.

Those modules of $\mathcal{W}(\mathfrak{gl}_r)$-algebras that take the form of higher quantum Airy structures were constructed in [4] by restricting twisted modules for the Heisenberg VOA associated to the Cartan subalgebra $\mathfrak{h}$ of $\mathfrak{gl}_r$ (see also [2, 3, 14] for related work). As such, the construction relies on a choice of automorphism $\sigma$ of $\mathfrak{h}$. In principle, arbitrary automorphisms can be considered. But it is not *a priori* obvious that the resulting modules may take the form of quantum higher Airy structures. While arbitrary automorphisms were briefly considered in [4], the paper mostly focused on the case where $\sigma$ is either the automorphism induced by the Coxeter element of the Weyl group (*i.e.* it permutes all basis vectors of $\mathfrak{h}$ cyclically) – see Theorem 4.9 in [4] – or the case where $\sigma$ permuted all but one basis vectors of $\mathfrak{h}$ – see Theorem 4.16. From the point of view of enumerative geometry, the former are related to various flavours of (closed) intersection theory on $\overline{\mathcal{M}}_{g,n}$ (or variants thereof), while the latter are related to open intersection theory.

A natural question then is to attempt to classify all higher quantum Airy structures that arise as modules of $\mathcal{W}(\mathfrak{gl}_r)$-algebras via the construction above, for abitrary automorphisms $\sigma$. This is the main motivation behind the current paper. At this stage, a full classification remains out of reach. But we make a small step in this direction in Section 3. We classify all higher quantum Airy structures that arise from automorphisms $\sigma$ consisting of products of disjoint cycles of the same length. This is achieved in Theorem 3.5.

One byproduct of Theorem 3.5 as that the dilaton shifts cannot simply be 1 anymore, as in [4]. In Section 3.5 we study a particularly interesting class of examples of the quantum $r$-Airy structures constructed in Theorem 3.5, where the dilaton shifts are taken to be roots of unity. This is perhaps the most natural way to achieve the invertibility requirement stated in Theorem 3.5.

It is also interesting to investigate the connection of these higher quantum Airy structures with topological recursion. The generalized topological recursion of [7–9] is a special case of higher quantum Airy structures for $\mathcal{W}(\mathfrak{gl}_r)$ obtained by taking $\sigma$ to be the automorphism induced by the Coxeter element of the Weyl group. What should the higher quantum Airy structures constructed from other choices of automorphisms correspond to? The natural guess is that they should produce a further generalization of topological recursion for spectral curves that are formulated as reducible algebraic curves. While we do not pursue this research direction in detail in this paper, we briefly explore this potential interpretation in Section 3.6, focusing on the higher quantum Airy structures obtained in Section 3.5 with dilaton shifts being roots of unity. It should be possible to use these higher quantum Airy structures to formulate such a generalization of topological recursion in terms of residue analysis on reducible spectral curves. We hope to probe this research direction further in the near future.

We also study whether the idea of Theorem 4.16 of [4] can be generalized further. In essence, what Theorem 4.16 is saying is that, given a quantum $r$-Airy structure constructed as a $\mathcal{W}(\mathfrak{gl}_r)$-module for $\sigma$ the fully cyclic automorphism (Theorem 4.9), one can always construct a new quantum $(r+1)$-Airy structure as a $\mathcal{W}(\mathfrak{gl}_{r+1})$-module by "appending" to $\sigma$ a trivial one-cycle, with no extra dilaton shift. As the higher quantum Airy structures of Theorem 4.9 are related to closed intersection theory, and those of Theorem 4.16 to open intersection theory, this could be understood as some sort of open/closed correspondence.

In Section 4 we prove under which conditions this idea of "appending a one-cycle" works. Namely, we start with any quantum $r$-Airy structure constructed from an arbitrary automorphism $\sigma$, and we prove under which condition appending a one-cycle, with no extra dilaton shift, gives rise to a new quantum $(r+1)$-Airy structure (see Theorem 4.1). While the enumerative geometric side of the story is unknown at this stage, if these arbitrary quantum $r$-Airy structures do have an interpretation in terms of closed intersection theory, the new ones may be expected to provide an open version of the enumerative geometric problem.

We conclude with future directions in Section 5. In particular, it would be very interesting to investigate whether these higher quantum Airy structures for more general automorphisms $\sigma$ have an enumerative geometric interpretation, and whether "appending one-cycles" as in Section 4 leads to a general open/closed correspondence statement.

**Remark**

Shortly after this paper was submitted to arXiv, a paper by Borot, Kramer and Schüler was submitted to arXiv [6], in which they also study the classification of higher quantum Airy structures obtained as modules of $\mathcal{W}(\mathfrak{gl}_{r+1})$-algebras. Their classification goes beyond the one presented in the current paper. They also propose a precise formulation of the corresponding topological recursion on reducible spectral curves that we discuss in Section 3.6.

## 2 Background

In this section we review the concept of higher quantum Airy structures, and the construction of [4], in order to motivate the problem addressed in this paper. We note that this paper is not meant to be self-contained: we simply recall the definitions and results that are necessary for the remainder of the paper. For the detailed construction of higher quantum Airy structures as modules of $\mathcal{W}$-algebras, the reader should consult [4].

### 2.1 Higher quantum Airy structures

We start by defining higher quantum Airy structures. For simplicity, we only state the explicit, basis-dependent definition:

**Definition 2.1** ([4], Definition 2.6)**.** Let $V$ be a vector space over $\mathbb{C}$. Let $(e_i)_{i \in I}$ be a basis of $V$, and $(x_i)_{i \in I}$ the corresponding basis of linear coordinates, where $I$ is an appropriate index set. Let $\hbar$ denote a formal parameter. Let $\mathcal{D}^{\hbar}$ be the completed algebra of differential operators on $V$, and define an algebra grading by assigning

$$\deg x_l = \deg \hbar \partial_{x_l} = 1, \qquad \deg \hbar = 2. \tag{1}$$

A *higher quantum Airy structure* on $V$ is a family of differential operators $(H_k)_{k \in I}$ such that:

1. They take the form

$$H_k = \hbar \partial_{x_k} - P_k, \tag{2}$$

   where $P_k \in \mathcal{D}^{\hbar}$ is a sum of terms of degree $\geq 2$ (this is a condition on the degree 0 and 1 terms of the $H_k$).

2. There exist $g^{k_3}_{k_1, k_2} \in \mathcal{D}^{\hbar}$ such that

$$\left[H_{k_1}, H_{k_2}\right] = \hbar \sum_{k_3 \in I} g^{k_3}_{k_1, k_2} H_{k_3}. \tag{3}$$

We define a *quantum r-Airy structure* as a higher quantum Airy structure such that all $P_k$ only have terms of degree $\leq r$.

### 2.2 Higher quantum Airy structures as modules of $\mathcal{W}(\mathfrak{gl}_r)$-algebras

In this paper we focus on higher quantum Airy structures that are obtained as modules of $\mathcal{W}(\mathfrak{gl}_r)$-algebras, following the construction of [4]. We will not go through the details of this construction here, but simply highlight its main features. The reader should supplement this paper with a careful reading of [4].

Before we describe these higher quantum Airy structures we make the following combinatorial definition.

**Definition 2.2.** For $r \in \mathbb{Z}^+$ we define a *λ-good index set*

$$\Lambda_r := \left\{ (i, m) \in \mathbb{Z}^2_{\geq 0} : 1 \leq i \leq r, \ m \geq i - \lambda(i) \right\}, \tag{4}$$

where we have defined

$$\lambda(i) = \min\{s | \lambda_1 + \cdots + \lambda_s \geq i\},$$

and $\lambda = (\lambda_1, \lambda_2, \ldots, \lambda_l)$ with $\lambda_1 \geq \lambda_2 \geq \ldots \geq \lambda_l \geq 1$ is an integer partition of $r$.

With this definition out of the way, we can highlight the main features of the construction of [4]. The starting point is the realization that the $\mathcal{W}(\mathfrak{gl}_r)$-algebra with central charge $c = r$ is strongly freely generated by exactly $r$ vectors $|v_i\rangle$, $i = 1, 2, \ldots, r$, in the Heisenberg VOA associated to the Cartan subalgebra $\mathfrak{h}$ of $\mathfrak{gl}_r$, with conformal weights $1, 2, \ldots, r$. The idea then is to construct a module $Y$ for the $\mathcal{W}(\mathfrak{gl}_r)$-algebra such that an appropriate subset of the modes of the fields $H^i(z) = Y(|v_i\rangle, z)$ takes the form of a higher quantum Airy structure.

More precisely, the construction is carried through the following steps (see Section 4 in [4]). Let $\mathfrak{h}$ be the Cartan subalgebra of $\mathfrak{gl}_r$, and $\sigma$ be an element of the Weyl group of $\mathfrak{gl}_r$. Let $|v_i\rangle$, $i = 1, 2, \ldots, r$ be the strong, free, generators of the $\mathcal{W}(\mathfrak{gl}_r)$-algebra with central charge $c = r$.

1. We construct a $\sigma$-twisted module $\mathcal{T}$ of the Heisenberg VOA associated to $\mathfrak{h}$. Upon restriction to the $\mathcal{W}(\mathfrak{gl}_r)$-algebra (which is a sub-VOA of the Heisenberg VOA), the module becomes untwisted. The underlying vector space of $\mathcal{T}$ is the space of formal series in countably many variables $x_1, x_2, \ldots$, and elements of $\mathcal{W}(\mathfrak{gl}_r)$ act as differential operators (of order at most $\mathrm{rank}(\mathfrak{gl}_r) = r$) in the $x_k$s.

2. We denote by $W^i(z) = \mathcal{T}(|v_i\rangle, z)$ the fields of the strong, free, generators of the $\mathcal{W}(\mathfrak{gl}_r)$-algebra. We pick a subset of the modes $W_m^i$ of these fields, such that $(i, m) \in \Lambda_r$ for some partition $\lambda$ of $r$. We call such a subset $\lambda$-good. It is shown in Section 3.3 of [4] that a subset of the modes $W_m^i$ fulfils condition (2) in the definition of a higher quantum Airy structure (Definition 2.1) if and only if it is $\lambda$-good for some partition $\lambda$ of $r$.

3. For the modes $W_m^i$ to form a higher quantum Airy structure, they must also satisfy condition (1) in Definition 2.1. This can potentially be achieved by conjugation (the so-called *dilaton shift*):

$$H_m^i = \hat{T}_s W_m^i \hat{T}_s^{-1}, \qquad \hat{T}_s := \exp\left(-\frac{Q \partial_{x_s}}{s}\right), \tag{5}$$

for some integer $s$ and constant $Q$, in conjunction with potential linear combinations of modes. (Note that by the Baker-Campbell-Hausdorff formula, (5) is equivalent to the shift

$$x_s \mapsto x_s - \frac{Q}{s}, \tag{6}$$

in the modes $W_m^i$.) If condition (1) can be achieved in this way, the $H_m^i$ form a quantum $r$-Airy structure, with the index set $I = \Lambda_r$ for the chosen partition $\lambda$ of $r$.

This construction was carried out in Section 4.1 of [4] for $\sigma$ the automorphism of the Cartan subalgebra $\mathfrak{h}$ of $\mathfrak{gl}_r$ induced by the Coxeter element of the Weyl group, which permutes cyclically all $r$ basis vectors of $\mathfrak{h}$. In this context, the main result is Theorem 4.9 of [4], which states that, for a given $r$, the construction above does produce a unique quantum $r$-Airy structure for each choice of integer $s \in \{1, 2, \ldots, r + 1\}$ such that $r = \pm 1 \bmod s$. The partition $\lambda$ defining the appropriate subalgebra of modes is uniquely fixed by the choice of $s$. See Theorem 4.9 in [4] for details.

But there is no reason *a priori* to focus on the automorphism $\sigma$ induced by the Coxeter element of the Weyl group: one could start with any automorphism $\sigma$ of the Weyl group. As an example, a more general case is studied in Section 4.2.2 of [4], where $\sigma$ is chosen to permute the $r - 1$ first basis vectors of $\mathfrak{h}$ and leave the last one invariant. The result is Theorem 4.16, which states that, for a given $r$, the construction does again produce a unique quantum $r$-Airy structure, but this time for each choice of integer $s \in \{1, \ldots, r\}$ such that $s | r$. As before, the partition $\lambda$ is uniquely fixed by the choice of $s$. In this case however, a new subtlety arises: one must consider linear combinations of the conjugated modes $H_m^i$ to ensure

that the condition (1) in Definition 2.1 is satisfied. But for this particular choice of $\sigma$, this can be achieved fairly easily (see Theorem 4.16).

One may then ask the following questions. Does the construction outlined above produce quantum $r$-Airy structures for all choices of automorphisms $\sigma$ of the Weyl group? And if so, for what choices of integer $s$? And, given a choice of $\sigma$ and $s$, is the corresponding partition $\lambda$ uniquely fixed?

In other words:

> Can one classify all quantum $r$-Airy structures that can be produced as modules of $\mathcal{W}(\mathfrak{gl}_r)$-algebras via the construction above?

It turns out that the main difficulty in producing such a classification lies in step (3). It is straightforward to construct the $\sigma$-twisted module (and its restriction to the $\mathcal{W}(\mathfrak{gl}_r)$-algebra) in step (1) for an arbitrary automorphism $\sigma$: in fact, this is already done in Section 4.2.1 of [4]. As for step (2), the classification of the subsets of modes that satisfy condition (2) in Definition 2.1) is already completed, as it is a purely algebraic property: it does not depend on the particular choice of $\mathcal{W}(\mathfrak{gl}_r)$-module. As mentioned above, the result is that the subset of modes satisfies condition (2) if and only if it is $\lambda$-good with respect to some partition $\lambda$ of $r$. What is tricky is to show that we can bring all modes $W_m^i$ in a chosen $\lambda$-good subset in a form that satisfies the condition (1) of Definition 2.1 via conjugation and linear combinations, *i.e.* step (3). This is rather non-trivial.

We can be a little more explicit. One can think of condition (1) of Definition 2.1 as having three parts:

(a) All operators have no degree 0 terms;

(b) All operators have no degree 1 terms that are coordinates $x_k$s;

(c) The degree one terms are all of the form $\hbar \partial_k$, and all derivatives $\hbar \partial_k$ appear exactly once in the degree 1 term of an operator.

If we have achieved conditions (a) and (b) by conjugation of the modes $W_m^i$, then what remains to be checked is that condition (c) can be satisfied by taking linear combinations of the conjugated modes. If the algebra was finitely generated, then this problem would be equivalent to the problem of determining invertibility of a finite-dimensional matrix. However, the subsets of modes that we are considering are infinite-dimensional. We are thus faced with the problem of determining invertibility of an infinite-dimensional matrix (via countably infinite elementary row operations).

The problem of inverting infinite-dimensional matrices is in general quite difficult. However, if the matrix is block-diagonal, then we may invert it if and only if the blocks are invertible, which drastically simplifies the problem.

In this paper we provide a classification of quantum $r$-Airy structures that can be obtained via the method above for a class of automorphisms $\sigma$ such that the resulting invertibility problem is block-diagonal. More specifically, we consider the case where $\sigma \in S_r$ is an automorphism of $\mathfrak{h}$ which is a product of disjoint cycles of the same length, and classify all resulting modules that take the form of quantum $r$-Airy structures. We also generalize Theorem 4.16 of [4], by studying under which conditions higher quantum Airy structures constructed from arbitrary automorphisms do produce new higher quantum Airy structures by "appending a one-cycle" with no extra dilaton shift.

However, a full classification of quantum $r$-Airy structures obtained as modules of $\mathcal{W}(\mathfrak{gl}_r)$-algebra via the recipe above for arbitrary automorphism $\sigma$ remains out of reach for the moment being. We hope to come back to this in the near future.

# 3  Higher quantum Airy structures for $\sigma$ a product of disjoint cycles of the same length

In this section we provide a classification of higher quantum Airy structures that can be obtained as modules of $\mathcal{W}(\mathfrak{gl}_r)$-algebras via the construction of the previous section, with the automorphism $\sigma \in S_r$ consisting of products of disjoint cycles of the same length. For our purposes, only the cycle structure of $\sigma$ matters.

We make heavy use of the construction of [4]. In particular, Lemma 4.15 in Section 4.2.1, which expresses the modes $W_m^i$ of the fields associated to the generators of the $\mathcal{W}(\mathfrak{gl}_r)$-algebra in terms of the Heisenberg modes, is our starting point.

## 3.1  Notation and previous results

Let us start by fixing notation. We consider $\mathcal{W}(\mathfrak{gl}_r)$. We write $\sigma = \prod_{j=1}^n \sigma_j \in S_r$ for the automorphism used to construct the twisted module of the underlying Heisenberg VOA, where each $\sigma_j$ is a cycle of length $\rho$, with $n\rho = r$.

In the construction of [4], there is a set of bosonic modes associated to each cycle of $\sigma$, and a corresponding set of coordinates. We denote by $K_m^j$, $j \in \{1, \ldots, n\}$, $m \in \mathbb{Z}$, the bosonic modes associated to the cycle $\sigma_j$, and we introduce the quantization

$$K_0^j = \hbar^{1/2} C_j, \qquad K_m^j = \hbar \partial_{x_m^j}, \qquad K_{-m}^j = m x_m^j, \qquad m \in \mathbb{Z}^+, \tag{7}$$

where the $C^j$ are constants (see Remark 4.14 in [4] for the appearance of the factor of $\hbar^{1/2}$).

Lemma 4.15 in [4] gives an explicit expression for the modes $W_m^i$, $i \in \{1, \ldots, r\}$, $m \in \mathbb{Z}$, of the fields associated to the generators of the $\mathcal{W}(\mathfrak{gl}_r)$-algebra in terms of the Heisenberg modes, as a result of the outlined construction for arbitrary automorphisms $\sigma$. For our choice of automorphism $\sigma = \prod_{j=1}^n \sigma_j$, the modes take the form

$$W_m^i = \frac{1}{\rho^i} \sum_{M \subseteq \{1,\ldots,n\}} \rho^{|M|} \sum_{\substack{1 \le i_j \le \rho, \ j \in M \\ \sum_{j \in M} i_j = i}} \sum_{\substack{m_j \in \mathbb{Z}, \ j \in M \\ \sum_j m_j = m+1-|M|}} \prod_{j \in M} W_{m_j}^{j,i_j}, \tag{8}$$

where the $W_{m_j}^{j,i_j}$, $j \in \{1, \ldots, n\}$, $i_j \in \{1, \ldots, \rho\}$, $m_j \in \mathbb{Z}$, are the modes of the $\mathcal{W}(\mathfrak{gl}_\rho)$-module constructed from the automorphism $\sigma_j$ induced by the Coxeter element of the Weyl group. Those are written in terms of the bosonic modes $K_m^j$ as:

$$W_m^{j,i} = \frac{1}{\rho} \sum_{\ell=0}^{\lfloor i/2 \rfloor} \frac{i! \hbar^\ell}{2^\ell \ell! (i-2\ell)!} \sum_{\substack{p_{2\ell+1},\ldots,p_i \in \mathbb{Z} \\ \sum_k p_k = \rho(m-i+1)}} \Psi_\rho^{(\ell)}(p_{2\ell+1}, \ldots, p_i) : \prod_{k=2\ell+1}^i K_{p_k}^j : . \tag{9}$$

We refer the reader to Definition 4.3 of [4] (and Section 4.2.1) for the definition of the $\Psi_\rho^{(\ell)}$.

## 3.2  Dilaton shifts

Equations (8) and (9) together express the modes $W_m^i$ in terms of the bosonic modes $K_m^j$. What remains to be shown is that we can find dilaton shifts, and possible linear combinations of modes, so that there exists a $\lambda$-good subset of modes (for some partition $\lambda$ of $r$) that satisfies condition (1) of Definition 2.1 in order to be a quantum higher Airy structure. For simplicity we will restrict to the case where we apply the same dilaton shift to all sets of bosonic modes associated to the $n$ cycles $\sigma_j$ of the automorphisms $\sigma$.

First, we recall from the proofs of Theorems 4.9 and 4.16 in [4] that if we do the dilaton shift

$$K^j_{-s} \mapsto K^j_{-s} - Q_j \tag{10}$$

in the modes $W^{j,i}_m$ of the $\mathcal{W}(\mathfrak{gl}_\rho)$-module constructed from $\sigma_j$, we obtain the resulting operators

$$H^{j,i}_m = -Q^i_j \delta_{\rho(m-i+1)+si} + Q^{i-1}_j K^j_{\rho m - (\rho-s)(i-1)} + \mathcal{O}(2), \tag{11}$$

where by $\mathcal{O}(2)$ we mean terms of order $\geq 2$ according to the algebra grading (1). The first term is of degree zero, while the second term is of degree one.

We may already restrict to the case where $s$ is coprime with $\rho$.[1] Indeed, let $d = \text{GCD}(\rho, s)$. Then the only modes $K^j_q$ that appear in the degree one terms of $H^{j,i}_m$ have $q$ divisible by $d$. So it will never be possible to achieve the degree condition (2) for a quantum $r$-Airy structure if $s$ is not coprime with $\rho$, as some derivatives $\hbar \partial_{x^j_m}$ will never appear in the degree one terms.

We thus assume from now on, without loss of generality, that $s$ is coprime with $\rho$. This implies that the degree zero term will be non-zero if and only if $i = \rho$ and $m = \rho - s - 1$. So we can rewrite the shifted operators as

$$H^{j,i}_m = -Q^i_j \delta_{i,\rho} \delta_{m,\rho-s-1} + Q^{i-1}_j K^j_{\rho m - (\rho-s)(i-1)} + \mathcal{O}(2). \tag{12}$$

With this under our belt we can prove the following lemma.

**Lemma 3.1.** *Consider the modes $W^i_m$, $i \in \{1, \ldots, r\}$, $m \in \mathbb{Z}$, of the $\mathcal{W}(\mathfrak{gl}_r)$-module constructed from an automorphism $\sigma = \prod^n_{j=1} \sigma_j$, where all $\sigma_j$ are disjoint cycles of length $\rho$, with $n\rho = r$. The expressions for the $W^i_m$ in terms of the bosonic modes are given by (8) and (9).*

*Apply the same dilaton shifts*

$$K^j_{-s} \to K^j_{-s} - Q_j, \qquad j \in \{1, \ldots, n\}, \tag{13}$$

*for all sets of bosonic modes on the operators $W^i_m$ to get new operators $H^i_m$. Here the $Q_j$ are some (potentially zero) constants, with $s$ an integer coprime with $\rho$. Then the resulting operators take the form:*

$$
H^{k+l\rho}_m = \frac{1}{\rho^{k+l\rho-l-1}} \left[ \delta_{k,\rho} \delta_{m,(l+1)(\rho-s)-1} \left( \sum_{\substack{M \subseteq \{1,\ldots,n\} \\ |M|=l+1}} \prod_{j \in M} (-Q^\rho_j) \right) \right.
$$
$$
\left. + \sum^n_{\mu=1} K^\mu_{\rho(m-l(\rho-s))-(\rho-s)(k-1)} Q^{k-1}_\mu \left( \sum_{\substack{M \subseteq \{1,\ldots,\hat{\mu},\ldots,n\} \\ |M|=l}} \prod_{j \in M} (-Q^\rho_j) \right) \right]
$$
$$
+ \mathcal{O}(2), \tag{14}
$$

*where $k \in \{1, 2, \ldots, \rho\}$, $l \in \{0, 1, \ldots, n-1\}$, $m \in \mathbb{Z}$, and we used the standard notation that $\{1, \ldots, \hat{\mu}, \ldots, n\}$ stands for the set $\{1, \ldots, n\}$ with the number $\mu$ omitted.*

*Proof.* The operators resulting from the chosen dilaton shifts are found by replacing the $W^{j,i_j}_{m_j}$ in (8) by the dilaton-shifted modes $H^{j,i_j}_{m_j}$ in (12). The result is:

$$
H^i_m = \frac{1}{\rho^i} \sum_{M \subseteq \{1,\ldots,n\}} \rho^{|M|} \sum_{\substack{1 \leq i_j \leq \rho, \, j \in M \\ \sum_{j \in M} i_j = i}} \sum_{\substack{m_j \in \mathbb{Z}, \, j \in M \\ \sum_j m_j = m+1-|M|}}
$$
$$
\times \prod_{j \in M} \left( -Q^{i_j}_j \delta_{i_j,\rho} \delta_{m_j,\rho-s-1} + Q^{i_j-1}_j K^j_{\rho m_j - (\rho-s)(i_j-1)} + \mathcal{O}(2) \right). \tag{15}
$$

---

[1] Of course, in the case with $\rho = 1$, this is trivial, as all integers $s$ are coprime with 1.

We are interested in the degree zero and degree one terms in $H_m^i$.

The degree zero term will appear when all factors in the product over $j \in M$ in (15) contribute a degree zero term. This will happen if and only if $i_j = \rho$ and $m_j = \rho - s - 1$ for all $j \in M$. Since $\sum_j i_j = i$, we conclude that this will happen only if $i = |M|\rho$ for some integer $|M|$ between 1 and $n$. Furthermore, since $\sum_j m_j = m + 1 - |M|$, we obtain that this will happen only if $m = |M|(\rho - s) - 1$.

It is thus appropriate to reindex the modes $H_m^i$ as $H_m^{k+l\rho}$, with $1 \le k \le \rho$ and $0 \le l \le n-1$. We then obtain

$$H_m^{k+l\rho} = \frac{1}{\rho^{(l+1)(\rho-1)}} \delta_{k,\rho} \delta_{m,(l+1)(\rho-s)-1} \left( \sum_{\substack{M \subseteq \{1,\dots,n\} \\ |M|=l+1}} \prod_{j \in M} (-Q_j^\rho) \right) + \mathcal{O}(1). \tag{16}$$

Next, we need to figure out the degree one terms. Degree one terms will appear when all factors in the product over $j \in M$ in (15) but one contribute a degree zero term. Let $\mu \in M$, and suppose that all terms with $j \in M$ and $j \ne \mu$ contribute a degree zero term. Thus $i_j = \rho$ and $m_j = \rho - s - 1$ for all $j \ne \mu$. We use the notation $i = k + l\rho$ as above to index the modes. Since $\sum_j i_j = k + l\rho$, we conclude that $i_\mu = k + l\rho - (|M|-1)\rho$. But $i_\mu$ must satisfy $1 \le i_\mu \le \rho$: we conclude that $|M| = l + 1$, and hence $i_\mu = k$. Furthermore, since $\sum_j m_j = m + 1 - |M|$, we conclude that $m_\mu = m - l(\rho - s)$. Putting all this together, we obtain:

$$\begin{aligned} H_m^{k+l\rho} = \frac{1}{\rho^{k+l\rho-l-1}} \Bigg[ &\delta_{k,\rho} \delta_{m,(l+1)(\rho-s)-1} \left( \sum_{\substack{M \subseteq \{1,\dots,n\} \\ |M|=l+1}} \prod_{j \in M} (-Q_j^\rho) \right) \\ &+ \sum_{\mu=1}^n K_{\rho(m-l(\rho-s))-(\rho-s)(k-1)}^\mu Q_\mu^{k-1} \left( \sum_{\substack{M \subseteq \{1,\dots,\hat{\mu},\dots,n\} \\ |M|=l}} \prod_{j \in M} (-Q_j^\rho) \right) \Bigg] \\ &+ \mathcal{O}(2). \end{aligned} \tag{17}$$

$\square$

A direct corollary of this lemma is the following:

**Corollary 3.2.** *Consider the modes $H_m^{k+l\rho}$, $k \in \{1,\dots,\rho\}$, $l \in \{0,1,\dots,n-1\}$, $m \in \mathbb{Z}$ from Lemma 3.1. If we restrict to the subset of modes $H_m^{k+l\rho}$ with*

$$m \ge l(\rho - s) + k - 1 - \left\lfloor \frac{s}{\rho}(k-1) \right\rfloor + \delta_{k,1}, \qquad s \ge 1, \tag{18}$$

*then the $H_m^{k+l\rho}$ have no degree zero terms, and the degree one terms all involve only bosonic modes $K_j^\mu$ with $j \ge 1$ (i.e. only derivatives $\hbar \partial_{x_j^\mu}$).*

*Proof.* This follows by direct inspection of (14). $\square$

We will focus on subsets of modes satisfying this condition from now on. We record for future use the form of the modes in this case, without the degree zero terms:

$$H_m^{k+l\rho} = \frac{1}{\rho^{k+l\rho-l-1}} \sum_{\mu=1}^n K_{\rho(m-l(\rho-s))-(\rho-s)(k-1)}^\mu Q_\mu^{k-1} \left( \sum_{\substack{M \subseteq \{1,\dots,\hat{\mu},\dots,n\} \\ |M|=l}} \prod_{j \in M} (-Q_j^\rho) \right) + \mathcal{O}(2). \tag{19}$$

What remains to be shown is twofold. First, that this subset of modes is a $\lambda$-good subalgebra, for some partition $\lambda$ of $r$ (so that condition (2) of Definition 2.1 is satisfied). Second, that there exist linear combinations of the $H_m^{k+l\rho}$ that satisfy condition (1) of Definition 2.1.

## 3.3 Linear combinations of operators

We address the second condition first. Looking at the linear terms in (19), we see that for a fixed value of $q$, the linear terms $K_q^\mu$, for $\mu = 1, \ldots, n$, all appear together in the same operators. This is key. What it means is that we are in a block-diagonal case. In other words, in order to show that there exist linear combinations of the operators $H_m^{k+l\rho}$ that satisfy condition (1) of Definition 2.1, all we have to do is determine whether the finite-dimensional matrices of coefficients corresponding to the block of modes where the $K_q^\mu$ appear (for fixed values of $q$) are invertible. The existence of a quantum $r$-Airy structure hinges on a block diagonal matrix inversion problem. This is what we now make rigorous.

**Definition 3.3.** Let $(Q_j)_{j=1}^n$ be a set of (possibly zero) constants, and let $\mu, \ell \in \{1, \ldots, n\}$. We define the $n$-by-$n$ *shift matrix*:

$$M(Q_1, \ldots, Q_n)_{\mu, \ell} = \sum_{\substack{M \subseteq \{1, \ldots, \hat{\mu}, \ldots, n\} \\ |M| = \ell - 1}} \prod_{j \in M} (-Q_j^\rho), \tag{20}$$

where we define $M(Q_1, \ldots, Q_n)_{\mu, 1} = 1$, so that if $n = 1$ we have $M(Q) = 1$ for any value of $Q$. We will use the shorthand notation $M_{\mu, \ell}$ when the dependence on the constants $Q_1, \ldots, Q_n$ is clear from context.

Using this definition, we can rewrite (19) as:

$$H_m^{k+l\rho} = \frac{1}{\rho^{k+l\rho-l-1}} \sum_{\mu=1}^n M_{\mu, l+1} Q_\mu^{k-1} K_{\rho(m-l(\rho-s))-(\rho-s)(k-1)}^\mu + \mathcal{O}(2). \tag{21}$$

This means that the block matrices in our block diagonal inversion problem are in fact all the same matrix $M$, with its $\mu$'th column multiplied by the constant $Q_\mu^{k-1}$. This means that for $Q_\mu^{k-1} \neq 0$ we have reduced the problem to the inversion of one finite-dimensional matrix.

More precisely:

**Lemma 3.4.** *Let $H_m^{k+l\rho}$, $k \in \{1, 2, \ldots, \rho\}$, $l \in \{0, 1, \ldots, n-1\}$, $m \in \mathbb{Z}$, be the operators constructed in Lemma 3.1. Consider the subalgebra of modes in Corollary 3.2, with*

$$m \geq l(\rho - s) + k - 1 - \left\lfloor \frac{s}{\rho}(k-1) \right\rfloor + \delta_{k,1}, \tag{22}$$

*where $s \geq 1$. Then there exist linear combinations of the operators $H_m^{k+l\rho}$ that satisfy condition (1) of Definition 2.1 if and only if the shift matrix $M(Q_1, \ldots, Q_n)$ is invertible, and either:*

*(a) $\rho = 1$;*

*(b) $\rho > 1$, $s$ is coprime with $\rho$, and $Q_j \neq 0$ for all $j = 1, \ldots, n$.*

*Proof.* We have already seen in Corollary 3.2 that the modes in the specified subset have no degree zero terms, and that the degree one terms are all derivatives $\hbar \partial_{x_m^\mu}$. What remains to be shown is that there exist linear combinations of the modes such that all derivative operators $\hbar \partial_{x_m^\mu}$ appear exactly once in the degree one terms.

Consider first the case $\rho = 1$. Then the operators read:

$$H_m^{1+l} = \sum_{\mu=1}^{n} M_{\mu,l+1} K_{m-l(1-s)}^{\mu} + \mathcal{O}(2), \tag{23}$$

with $m \geq l(1-s)+1$. We then notice that, for any fixed value of $q \geq 1$, the modes $K_q^{\mu}$ appear together in the $n$ modes $H_{q+l(1-s)}^{1+l}$, all of which are in the specified subset. For any $q$, the matrix of coefficients is precisely the shift matrix $M$. Therefore, there exist linear combinations of the operators such that all bosonic modes $K_q^{\mu}$ appear exactly once in the linear terms if and only if the shift matrix $M$ is invertible.

Now consider the case $\rho > 1$. The operators read:

$$H_m^{k+l\rho} = \frac{1}{\rho^{k+l\rho-l-1}} \sum_{\mu=1}^{n} M_{\mu,l+1} Q_{\mu}^{k-1} K_{\rho(m-l(\rho-s))-(\rho-s)(k-1)}^{\mu} + \mathcal{O}(2), \tag{24}$$

with

$$m \geq l(\rho-s)+k-1-\left\lfloor \frac{s}{\rho}(k-1) \right\rfloor + \delta_{k,1}. \tag{25}$$

An argument similar to the case $\rho = 1$ holds here as well. For any fixed value of $q \geq 1$, the modes $K_q^{\mu}$ appear together in exactly $n$ modes, with the matrix of coefficients given by $M_{\mu,l+1} Q_{\mu}^{k-1}$. Indeed, fix a $q \geq 1$, and consider the modes $H_m^k$ (with $l = 0$). Then, one can always find a unique choice of $m$ and $k$ such that the modes $K_q^{\mu}$ appear in the linear terms $H_m^k$ by solving the equation $\rho m - (\rho-s)(k-1) = q$, if and only if $s$ is coprime with $\rho$ (otherwise it would only be possible for a subset of $q$'s that are multiple of GCD$(\rho,s)$, see the discussion at the beginning of Section 3.2). But then the same modes $K_q^{\mu}$ also appear in the linear terms of the operators $H_{m+l(\rho-s)}^{k+l\rho}$ for all $l \in \{0, \ldots, n-1\}$. Furthermore, the inequality (25) is precisely such that all these modes are included in the specified subset. As a result, assuming that all $Q_{\mu} \neq 0$, we can always find linear combinations of the operators such that all bosonic modes $K_q^{\mu}$ appear exactly once in the linear terms if and only if the shift matrix $M$ is invertible and $s$ is coprime with $\rho$.

We assumed however that all $Q_{\mu} \neq 0$. Is that a necessary condition? What happens if some of the constants $Q_{\mu}$ vanish? Assume that there is a vanishing constant, which we take to be $Q_1$, without loss of generality. Then the operators with $k > 1$ become

$$H_m^{k+l\rho} = \frac{1}{\rho^{k+l\rho-l-1}} \sum_{\mu=2}^{n} M_{\mu,l+1} Q_{\mu}^{k-1} K_{\rho(m-l(\rho-s))-(\rho-s)(k-1)}^{\mu} + \mathcal{O}(2). \tag{26}$$

Thus the modes $K_q^1$ only appear in the operators $H_m^{1+l\rho}$ (with $k = 1$). But only the modes $K_q^1$ with $q$ divisible by $\rho$ appear in these operators. Therefore, in the case $\rho > 1$, it is necessary for all the $Q_{\mu}$ to be non-zero for all the bosonic modes to appear in the linear terms. $\qquad \square$

## 3.4 Classification Theorem

All that remains is to determine whether the specified subsets are $\lambda$-good for some choice of integer partition $\lambda$ of $r$. The result is the main theorem of this section. We only consider the case with $n \geq 2$ (i.e. automorphisms $\sigma$ with more than one cycle), as the case $n = 1$ corresponds to the quantum $r$-Airy structures constructed in Theorem 4.9 of [4].

**Theorem 3.5.** *Let $H_m^{k+l\rho}$, $k \in \{1, 2, \ldots, \rho\}$, $l \in \{0, 1, \ldots, n-1\}$, $m \in \mathbb{Z}$, $n \geq 2$, and $r = n\rho$, be the operators constructed in Lemma 3.1 (that is, they are constructed as restrictions of twisted*

modules of the Heisenberg algebra, where the twist is given by the automorphism $\sigma = \prod_{j=1}^{n} \sigma_j$ with the $\sigma_j$ disjoint cycles of length $\rho$). Consider the subset of modes in Corollary 3.2, with

$$m \geq l(\rho - s) + k - 1 - \left\lfloor \frac{s}{\rho}(k-1) \right\rfloor + \delta_{k,1} \,, \tag{27}$$

where $s \geq 1$. Then there exist linear combinations of the operators $H_m^{k+l\rho}$ that form a quantum $r$-Airy structure if and only if $\sum_{j=1}^{n} K_0^j = 0$, the shift matrix $M(Q_1, \ldots, Q_n)$ is invertible, and one of the following conditions is satisfied:

(a) $\rho = 1$, $s = 1$, any number $n$ of 1-cycles;

(b) $\rho > 1$, $s = 1$, any number $n$ of $\rho$-cycles, and $Q_j \neq 0$ for all $j = 1, \ldots, n$;

(c) $\rho = 1$, $s = 2$, $n = 2$ (two 1-cycles);

(d) $\rho > 1$, $\rho$ is odd, $s = 2$, $n = 2$ (two $\rho$-cycles), and $Q_j \neq 0$ for all $j = 1, \ldots, n$.

**Remark 3.6.** Before we prove this classification theorem, let us mention that the result is perhaps a little bit unexpected. In the case with $n = 1$ considered in Section 4.1 of [4], there are much larger choices of $s$ that give rise to quantum $r$-Airy structures, namely any $s \in \{1, 2, \ldots, r+1\}$ such that $r = \pm 1 \mod s$. One could have expected a similar range of possibilities here. However, it appears to be much more constrained when $n \geq 2$. Basically, the operators can form a quantum $r$-Airy structure only for $s = 1$, with the exception of the case with $n = 2$ (*i.e.* two cycles) where $s = 2$ can also work.

*Proof.* We first prove that cases (a)-(d) form quantum $r$-Airy structures. We then prove that these are the only possibilities.

In all cases (a)-(d), we know from Lemma 3.4 that condition (1) of Definition 2.1 is satisfied. What we need to check is that the subset of modes is $\lambda$-good for some choice of partition $\lambda$ of $r$, from which we can conclude that condition (2) of Definition 2.1 is satisfied, and therefore that the operators form a quantum $r$-Airy structure.

(a) For $\rho = 1$ and $s = 1$, we must have $k = 1$, and the operators read

$$H_m^{1+l} = \sum_{\mu=1}^{n} M_{\mu, l+1} K_m^\mu + \mathcal{O}(2) \,, \tag{28}$$

with $l \in \{0, 1, \ldots, n-1\}$, and

$$m \geq 1 \,. \tag{29}$$

We need to check whether this subset is $\lambda$-good for some partition $\lambda$ of $r = n$. It clearly is not, since any $\lambda$-good subset must include the operator $H_0^1$.[2] But if we add the operator $H_0^1$ to the subset, then it is straightforward to check that the subset is $\lambda$-good for the partition

$$\lambda = (2, 1, \ldots, 1) \tag{30}$$

of $r$, as it corresponds to the subset of modes given by

$$m \geq 1 + l - \lambda(1+l) = 1 - \delta_{l,0} \,. \tag{31}$$

However, the extra operator $H_0^1 = \sum_{\mu=1}^{n} K_0^\mu$ does not satisfy condition (1) of Definition 2.1, and hence we must require that $\sum_{\mu=1}^{n} K_0^\mu = 0$ so that the operator identically vanishes.

---

[2]Indeed, recall that for a partition $\lambda$ of $r$, the index set $\Lambda_r$ is defined as $\Lambda_r = \left\{ (i, m) \in \mathbb{Z}_{\geq 0}^2 : 1 \leq i \leq r, \ m \geq i - \lambda(i) \right\}$. In particular, since for any partition $\lambda(1) = 1$, we must have $(1, 0) \in \Lambda_r$, and hence $H_0^1$ must be in the $\lambda$-good subset.

(b) We now consider the case $\rho > 1$ and $s = 1$. The operators read:

$$H_m^{k+l\rho} = \frac{1}{\rho^{k+l\rho-l-1}} \sum_{\mu=1}^{n} M_{\mu,l+1} Q_\mu^{k-1} K_{\rho(m-l(\rho-1))-(\rho-1)(k-1)}^\mu + \mathcal{O}(2), \tag{32}$$

with

$$m \geq k + l\rho - (l+1) - \left\lfloor \frac{k-1}{\rho} \right\rfloor + \delta_{k,1} = k + l\rho - (l+1) + \delta_{k,1}, \tag{33}$$

where the equality follows since $k \in \{1, 2, \ldots, \rho\}$. We need to check whether this subset is $\lambda$-good for some partition $\lambda$ of $r$. As before, it clearly is not, since $H_0^1$ is not included. But after adding $H_0^1$ to the subset, it becomes $\lambda$-good with respect to the partition

$$\lambda = (\rho + 1, \rho, \ldots, \rho, \rho - 1) \tag{34}$$

of $r$. Indeed, for this partition we have $\lambda(k + l\rho) = l + 1 - \delta_{k,1} + \delta_{k,1}\delta_{l,0}$, and thus the $\lambda$-good subset of modes is given by

$$m \geq k + l\rho - \lambda(k + l\rho) = k + l\rho - (l+1) + \delta_{k,1} - \delta_{k,1}\delta_{l,0}. \tag{35}$$

As $H_0^1 = \sum_{\mu=1}^{n} K_0^\mu$ does not satisfy condition (1), we require that $\sum_{\mu=1}^{n} K_0^\mu = 0$ so that the operator identically vanishes.

(c) We now consider the case $\rho = 1$ and $s = 2$. Then $k = 1$, and the operators read:

$$H_m^{1+l} = \sum_{\mu=1}^{n} M_{\mu,l+1} K_{m+l}^\mu + \mathcal{O}(2), \tag{36}$$

with

$$m \geq -l + 1, \tag{37}$$

for $l \in \{0, \ldots, n-1\}$. To get a $\lambda$-good subset, all $m$ must be at least non-negative (see Section 3.3 of [4], for instance the beginning of the proof of Theorem 3.16, and also Proposition 3.14: the largest $\lambda$-good subset corresponds to the partition $\lambda = (r)$ of $r$, and it consists of all modes with $m \geq 0$). Thus we cannot get a $\lambda$-good subset for $n > 2$, as the specified subset contains negative modes, since $l \in \{0, \ldots, n-1\}$. Now consider $n = 2$. In this case, this is not quite a $\lambda$-good subset, but if we include, as usual, the mode $H_0^1$, we get the $\lambda$-good subset corresponding to the partition

$$\lambda = (2) \tag{38}$$

of $r = n = 2$. Thus we get a quantum 2-Airy structure if we impose that $H_0^1 = \sum_{\mu=1}^{2} K_0^\mu = 0$.

(d) We consider $\rho > 1$ and $s = 2$. We note that $\rho$ must be odd, as it is coprime with $s = 2$. The operators read:

$$H_m^{k+l\rho} = \frac{1}{\rho^{k+l\rho-l-1}} \sum_{\mu=1}^{n} M_{\mu,l+1} Q_\mu^{k-1} K_{\rho(m-l(\rho-2))-(\rho-2)(k-1)}^\mu + \mathcal{O}(2), \tag{39}$$

with

$$m \geq k + l\rho - (2l+1) - \left\lfloor \frac{2}{\rho}(k-1) \right\rfloor + \delta_{k,1}, \tag{40}$$

for $k \in \{1, 2, \ldots, \rho\}$ and $l \in \{0, \ldots, n-1\}$. We need to determine whether these subsets are $\lambda$-good.

Consider first the case $n = 2$. To get a $\lambda$-good subset, as usual we need to include the mode $H_0^1$. Thus we must require that $H_0^1 = \sum_{\mu=1}^{2} K_0^\mu = 0$. With this mode included, the subset is $\lambda$-good for the partition $\lambda$ of $r = 2\rho$ given by (recall that $\rho$ is odd):

$$\lambda = \left(\frac{\rho+1}{2}, \frac{\rho+1}{2}, \frac{\rho-1}{2}, \frac{\rho-1}{2}\right). \tag{41}$$

Indeed, for this partition $\lambda(k+l\rho) = 2l + 1 + \left\lfloor \frac{2}{\rho}(k-1) \right\rfloor - \delta_{k,1} + \delta_{k,1}\delta_{l,0}$, and hence the subset of modes is determined by the condition

$$m \geq k + l\rho - \lambda(k+l\rho) = k + l\rho - (2l+1) - \left\lfloor \frac{2}{\rho}(k-1) \right\rfloor + \delta_{k,1} - \delta_{k,1}\delta_{l,0}. \tag{42}$$

For $n \geq 3$ however, one can show that there is no partition of $r$ that gives rise to the desired subset of modes. Indeed, what we are trying to find is a partition $\lambda = (\lambda_1, \ldots, \lambda_a)$ of $r$, with $\lambda_1 \geq \lambda_2 \geq \ldots \geq \lambda_a \geq 1$, such that

$$\lambda(k+l\rho) = 2l + 1 + \left\lfloor \frac{2}{\rho}(k-1) \right\rfloor - \delta_{k,1} + \delta_{k,1}\delta_{l,0}, \tag{43}$$

for $k \in \{1, 2, \ldots, \rho\}$ and $l \in \{0, \ldots, n-1\}$. We start with $l = 0$. For $k \in \{1, \ldots, \frac{\rho+1}{2}\}$, the condition is that $\lambda(k) = 1$. Continuing with $k \in \{\frac{\rho+3}{2}, \ldots, \rho\}$, we get $\lambda(k) = 2$. This tells us that the first part of the partition $\lambda$ should be $\lambda_1 = \frac{\rho+1}{2}$. Continuing with $l = 1$, for $k = 1$ we get $\lambda(1+\rho) = 2$, for $k \in \{2, \ldots, \frac{\rho+1}{2}\}$, we get $\lambda(k+\rho) = 3$, and for $k \in \{\frac{\rho+3}{2}, \ldots, \rho\}$ we get $\lambda(k+\rho) = 4$. This tells us that the second part of the partition $\lambda$ should be $\lambda_2 = \frac{\rho+1}{2}$, and the third part should be $\lambda_3 = \frac{\rho-1}{2}$. But then, continuing with $l = 2$, for $k = 1$ we get $\lambda(1+2\rho) = 4$, for $k \in \{2, \ldots, \frac{\rho+1}{2}\}$, we get $\lambda(k+2\rho) = 5$, and for $k \in \{\frac{\rho+3}{2}, \ldots, \rho\}$ we get $\lambda(k+2\rho) = 6$. This tells us that the fourth part of the partition $\lambda$ should be $\lambda_4 = \frac{\rho+1}{2}$, which is a contradiction, since $\lambda_4 \geq \lambda_3$. We conclude that the chosen subset is not $\lambda$-good for $n \geq 3$, and thus we cannot get a quantum $r$-Airy structure for $n \geq 3$; only the $n = 2$ case survives.

Now that we have proved that cases (a)-(d) form quantum $r$-Airy structures, what remains is to show that these are the only possibilities. In other words, we want to show that the specified subsets of modes for other choices of $s \geq 1$ are not $\lambda$-good.

First, from Lemma 3.4, we know that condition (1) of Definition 2.1 is satisfied if and only if the shift matrix $M$ is invertible, and either $\rho = 1$, or $\rho > 1$, in which case $s$ must be coprime with $\rho$ and $Q_j \neq 0$ for all $j = 1, \ldots, n$.

Consider first the case $\rho = 1$ (in which case $k = 1$), with a shift $s \geq 3$ coprime with $\rho$. The operators read

$$H_m^{1+l} = \sum_{\mu=1}^{n} M_{\mu,l+1} K_{m-l(1-s)}^\mu + \mathcal{O}(2), \tag{44}$$

with

$$m \geq 1 - l(s-1), \tag{45}$$

with $l \in \{0, 1, \ldots, n-1\}$. We know that all $m$ must be at least non-negative to get a $\lambda$-good subset (as mentioned before, see Section 3.3 of [4], in particular the beginning of the proof of Theorem 3.16 and Proposition 3.14). But since $n \geq 2$, this is impossible for $s \geq 3$ (even after adding the mode $H_0^1$ to the subalgebra). Therefore the only possible choices of $s$ are $s = 1, 2$, which were considered in cases (a) and (c).

Consider now the case $\rho > 1$, with a shift $s \geq 3$. The operators read

$$H_m^{k+l\rho} = \frac{1}{\rho^{k+l\rho-l-1}} \sum_{\mu=1}^{n} M_{\mu,l+1} Q_\mu^{k-1} K_{\rho(m-l(\rho-s))-(\rho-s)(k-1)}^{\mu} + \mathcal{O}(2), \tag{46}$$

with

$$m \geq l(\rho - s) + k - 1 - \left\lfloor \frac{s}{\rho}(k-1) \right\rfloor + \delta_{k,1}, \tag{47}$$

for $k \in \{1, 2, \ldots, \rho\}$ and $l \in \{0, 1, \ldots, n-1\}$.

Let us assume first that $s > \rho$. Then some of the modes in the subset (consider for instance $k = 2$ and $l = 1$) will have negative $m$'s. But as mentioned above, we know that all $m$ must be at least non-negative for the subset to potentially be $\lambda$-good, and thus we must have $s < \rho$.

The question then is: for $3 \leq s \leq \rho - 1$, can we find a partition $\lambda$ of $r = n\rho$ such that

$$\lambda(k+l\rho) = sl + 1 + \left\lfloor \frac{s}{\rho}(k-1) \right\rfloor - \delta_{k,1} + \delta_{k,1}\delta_{l,0} \ ? \tag{48}$$

We can try to build such a partition. After a tedious calculation similar to what we did above for case (d), we see that the "partition" would have to look like:

$$\lambda = (A_1, A_2, \ldots, A_{s-1}, A_s + 1, A_1 - 1, A_2, \ldots, A_{s-1}, \ldots), \tag{49}$$

where we defined

$$A_j = \left\lceil \frac{\rho j}{s} \right\rceil - \left\lceil \frac{\rho(j-1)}{s} \right\rceil. \tag{50}$$

But for this to be a partition, we must have that $\lambda_1 \geq \lambda_2 \geq \ldots \geq 1$. Since all $A_j$ with $j = 2, \ldots, s-1$ appear before and after $A_1 - 1$, we must have

$$A_2 = A_3 = \ldots = A_{s-1} = A_1 - 1 = a, \tag{51}$$

for some positive integer $a$, and then we must also have

$$A_s + 1 = a. \tag{52}$$

But by definition of the $A_j$, we have:

$$\sum_{j=1}^{s} A_j = \rho, \tag{53}$$

and thus we get:

$$as = \rho. \tag{54}$$

But $s$ is coprime with $\rho$, which is a contradiction. Therefore, there is no $\lambda$-good subset of operators for $s \geq 3$ and $\rho > 1$. The only possible choices are $s = 1$ and $s = 2$, which were considered in cases (b) and (d).

This completes the proof of the theorem. □

## 3.5 An interesting class of examples

An interesting feature of Theorem 3.5 is that the dilaton shifts $Q_j$, $j = 1, \ldots, n$, cannot be taken to be simply 1 anymore, in contrast to Theorem 4.9 in [4]. Indeed, the shift matrix $M(Q_1, \ldots, Q_n)$ must be invertible, which restricts possible choices of dilaton shifts.

There is however a natural way of ensuring invertibility of the shift matrix for all quantum $r$-Airy structures constructed in Theorem 3.5. The idea is to let $Q_j = \omega^j$, $j = 1, \ldots, n$, where $\omega$ is a primitve $r$'th root of unity (recall that $r = n\rho$). We study this interesting class of examples

in this section. They appear to be intimately connected to the geometry of reducible spectral curves, as we briefly explore in Section 3.6.

Let us start by proving a simple lemma about roots of unity, which will be necessary to prove invertibility of the shift matrix.

**Lemma 3.7.** *Let $n \in \mathbb{Z}^+$, with $n \geq 2$, and $\mu, \ell \in \{1, 2, \ldots, n\}$. Let $\theta$ be a primitive $n$-th root of unity. Then*

$$\sum_{\substack{M \subseteq \{1, \ldots, \hat{\mu}, \ldots, n\} \\ |M| = \ell - 1}} \prod_{j \in M} (-\theta^j) = \theta^{\mu(\ell-1)}. \tag{55}$$

*Proof.* Vieta's formula gives:

$$\sum_{\substack{M \subseteq \{1, \ldots, n\} \\ |M| = \ell - 1}} \prod_{j \in M} \theta^j = 0. \tag{56}$$

We can separate on the left-hand-side contributions from subsets that include the index $\mu \in \{1, \ldots, n\}$. Rearranging, we get:

$$\sum_{\substack{M \subseteq \{1, \ldots, \hat{\mu}, \ldots, n\} \\ |M| = \ell - 1}} \prod_{j \in M} \theta^j = -\theta^\mu \sum_{\substack{M \subseteq \{1, \ldots, \hat{\mu}, \ldots, n\} \\ |M| = \ell - 2}} \prod_{j \in M} \theta^j. \tag{57}$$

Doing this iteratively, we conclude that

$$\sum_{\substack{M \subseteq \{1, \ldots, \hat{\mu}, \ldots, n\} \\ |M| = \ell - 1}} \prod_{j \in M} \theta^j = (-1)^{\ell-1} \theta^{\mu(\ell-1)}, \tag{58}$$

from which the statement of the lemma follows. $\qquad\square$

An immediate corollary is the following:

**Corollary 3.8.** *Let $\rho, n \in \mathbb{Z}^+$ with $n \geq 2$, and $\mu, \ell \in \{1, 2, \ldots, n\}$. Let $r = n\rho$, and $\omega$ be a primitive $r$-th root of unity. Let $Q_j = \omega^j$ for $j = 1, 2, \ldots, n$, and define $\theta = \omega^\rho$, which is a primitive $n$-th root of unity. Then the shift matrix (see Definition 3.3)*

$$M(\omega, \omega^2, \ldots, \omega^n)_{\mu, \ell} = \sum_{\substack{M \subseteq \{1, \ldots, \hat{\mu}, \ldots, n\} \\ |M| = \ell - 1}} \prod_{j \in M} (-\omega^{\rho j}) = \omega^{\rho \mu(\ell-1)} = \theta^{\mu(\ell-1)}. \tag{59}$$

*Note that the shift matrix $M$ can be written explicitly as the $n \times n$ Vandermonde matrix:*

$$M = \begin{bmatrix} 1 & \theta^1 & \theta^2 & \ldots & \theta^{(n-1)} \\ 1 & \theta^2 & \theta^4 & \ldots & \theta^{2(n-1)} \\ \vdots & \vdots & \vdots & \ddots & \vdots \\ 1 & \theta^{(n-1)} & \theta^{2(n-1)} & \ldots & \theta^{(n-1)(n-1)} \\ 1 & 1 & 1 & \ldots & 1 \end{bmatrix}. \tag{60}$$

*In particular, it is invertible.*

The upshot is that we have constructed a large class of quantum $r$-Airy structures with interesting potential interpretations. Consider a quantum $r$-Airy structure constructed as in Theorem 3.5, that is, as a $\mathcal{W}(\mathfrak{gl}_r)$-module descending from a twisted module of the underlying Heisenberg algebra with the twist given by an automorphism $\sigma = \prod_{j=1}^n \sigma_j$, with each cycle $\sigma_j$ of length $\rho$. Here $r = n\rho$. Then choose the dilaton shifts $Q_j$, $j = 1, \ldots, n$ to be given by $Q_j = \omega^j$, where $\omega$ is a primitive $r$-th root of unity. This choice of dilaton shifts always

satisfy the invertibility condition in Theorem 3.5, and, assuming that we are in one of the cases specified in the theorem, we obtain a quantum $r$-Airy structure.

We will come back to a potentially interesting interpretation for this class of quantum $r$-Airy structures in Section 3.6. Meanwhile, let us write down in detail one of the simplest higher quantum Airy structures in this class, to make things explicit.

**Example 3.9.** We write down in detail the operators of the quantum 4-Airy structure obtained as a module of the $\mathcal{W}(\mathfrak{gl}_4)$-algebra via restriction of a twisted module of the underlying Heisenberg algebra, where the twist results from the automorphism $\sigma = \sigma_1 \sigma_2$ with two disjoint 2-cycles. We consider the case with $s = 1$, which is part of the family (b) in Theorem 3.5. To satisfy the invertibility condition of the shift matrix, we take the dilaton shifts $Q_j$, $j = 1, 2$ to be given by $Q_1 = i$, $Q_2 = i^2 = -1$, where $i = \sqrt{-1}$, as in Corollary 3.8.

We let $K_m^j$, $j = 1, 2$, be the bosonic modes associated to the two 2-cycles, and let $W_m^{j,i}$, $j = 1, 2$, $i = 1, 2$, be the modes of the two $\mathcal{W}(\mathfrak{gl}_2)$-modules associated to the cycles $\sigma_j$, $j = 1, 2$. The modes $W_m^i$, $i = 1, \ldots, 4$ of the resulting $\mathcal{W}(\mathfrak{gl}_4)$ can be written in terms of those as:

$$
\begin{aligned}
W_m^1 &= W_m^{1,1} + W_m^{2,1}, \\
W_m^2 &= \frac{1}{2} W_m^{1,2} + \frac{1}{2} W_m^{2,2} + \sum_{\substack{m_1, m_2 \in \mathbb{Z} \\ m_1 + m_2 = m-1}} W_{m_1}^{1,1} W_{m_2}^{2,1}, \\
W_m^3 &= \frac{1}{2} \sum_{\substack{m_1, m_2 \in \mathbb{Z} \\ m_1 + m_2 = m-1}} \left( W_{m_1}^{1,1} W_{m_2}^{2,2} + W_{m_1}^{1,2} W_{m_2}^{2,1} \right), \\
W_m^4 &= \frac{1}{4} \sum_{\substack{m_1, m_2 \in \mathbb{Z} \\ m_1 + m_2 = m-1}} W_{m_1}^{1,2} W_{m_2}^{2,2},
\end{aligned}
\tag{61}
$$

with the subalgebra condition $m \geq \left\lfloor \frac{i+1}{2} \right\rfloor$. We then implement the dilaton shifts:

$$
K_{-1}^j \mapsto K_{-1}^j - i^j, \qquad j = 1, 2. \tag{62}
$$

After the shift, the modes of the two $\mathcal{W}(\mathfrak{gl}_2)$-modules become:

$$
\begin{aligned}
H_m^{j,1} &= K_{2m}^j, \\
H_m^{j,2} &= i^j K_{2m-1}^j - \frac{(-1)^j}{2} \delta_{m,0} + \frac{1}{2} \sum_{\substack{p_1, p_2 \in \mathbb{Z} \\ p_1 + p_2 = 2(m-1)}} \left( 2\delta_{2|p_1} \delta_{2|p_2} - 1 \right) : K_{p_1}^j K_{p_2}^j : -\frac{3\hbar}{24} \delta_{m,1},
\end{aligned}
\tag{63}
$$

for $j = 1, 2$. Finally, replacing the $W_m^{j,i}$ in (61) by the shifted $H_m^{j,i}$, which implements the dilaton shifts on the $\mathcal{W}(\mathfrak{gl}_4)$-module, yields the operators of the resulting quantum 4-Airy structure. We write down explicitly the resulting expanded form of $H_m^1$, $H_m^2$, and $H_m^3$, but leave out $H_m^4$ for brevity, as its expanded form is rather long:

$$H_m^1 = K_{2m}^1 + K_{2m}^2 \,, \tag{64}$$

$$2H_m^2 = iK_{2m-1}^1 - K_{2m-1}^2 + \frac{1}{2}\sum_{\substack{p_1,p_2\in\mathbb{Z}\\p_1+p_2=2(m-1)}}\left(2\delta_{2|p_1}\delta_{2|p_2}-1\right)\left(:K_{p_1}^1K_{p_2}^1: + :K_{p_1}^2K_{p_2}^2:\right)$$

$$+ 2\sum_{\substack{m_1,m_2\in\mathbb{Z}\\m_1+m_2=m-1}}K_{2m_1}^1K_{2m_2}^2 - \frac{3\hbar}{12}\delta_{m,1}\,, \tag{65}$$

$$4H_m^3 = -K_{2m-2}^1 + K_{2m-2}^2 - \frac{3\hbar}{12}\left(K_{2m-4}^1 + K_{2m-4}^2\right)$$

$$- 2\sum_{\substack{m_1,m_2\in\mathbb{Z}\\m_1+m_2=m-1}}K_{2m_1}^1K_{2m_2-1}^2 + 2i\sum_{\substack{m_1,m_2\in\mathbb{Z}\\m_1+m_2=m-1}}K_{2m_1-1}^1K_{2m_2}^2$$

$$+ \sum_{\substack{m_1,m_2\in\mathbb{Z}\\m_1+m_2=m-1}}\sum_{\substack{p_1,p_2\in\mathbb{Z}\\p_1+p_2=2(m_2-1)}}\left(2\delta_{2|p_1}\delta_{2|p_2}-1\right)K_{2m_1}^1:K_{p_1}^2K_{p_2}^2:$$

$$+ \sum_{\substack{m_1,m_2\in\mathbb{Z}\\m_1+m_2=m-1}}\sum_{\substack{p_1,p_2\in\mathbb{Z}\\p_1+p_2=2(m_1-1)}}\left(2\delta_{2|p_1}\delta_{2|p_2}-1\right)K_{2m_2}^2:K_{p_1}^1K_{p_2}^1:\,, \tag{66}$$

$$8H_m^4 = -iK_{2m-3}^1 - K_{2m-3}^2 + \mathcal{O}(2)\,, \tag{67}$$

with the subalgebra of mode given by $m \geq \left\lfloor\frac{i+1}{2}\right\rfloor$. As required by Theorem 3.5, we also impose that

$$H_0^1 = K_0^1 + K_0^2 = 0\,, \tag{68}$$

but each $K_0^j$ does not have to vanish independently.

## 3.6 Higher quantum Airy structures and topological recursion

It is interesting to try to connect the construction of higher quantum Airy structures in Theorem 3.5 to the Chekhov, Eynard, and Orantin topological recursion. It is shown in [4] that the generalized topological recursion of [7–9] can be reformulated as a special case of higher quantum Airy structures realized as $\mathcal{W}(\mathfrak{gl}_r)$-modules, originating from twisted modules of the underlying Heisenberg algebra with the twist given by the automorphism induced by the Coxeter element of the Weyl group. Indeed, this was the original motivation for the study of higher quantum Airy structures [4]. Note that the original topological recursion of Chekhov, Eynard, and Orantin then corresponds to the special case of (quadratic) quantum Airy structures originally studied by Kontsevich and Soibelman [1, 13].

The higher quantum Airy structures that are relevant for the generalized topological recursion of [7–9] are those of Theorem 4.9 [4], which are indexed by an integer $r \geq 2$ and another integer $s \in \{1, \ldots, r+1\}$ such that $r = \pm 1 \bmod s$ (in particular, $r$ and $s$ are coprime). They are constructed as $\mathcal{W}(\mathfrak{gl}_r)$-modules, with dilaton shift

$$K_{-s} \mapsto K_{-s} - 1\,. \tag{69}$$

Recall that the topological recursion relies on the geometry of a spectral curve. It is shown in [4] that those quantum $r$-Airy structures encapsulate the same information as the topological recursion of [7–9] on the so-called $(r,s)$-spectral curves, which are realized as the algebraic curves

$$r^{r-s}x^{r-s}y^r - (-1)^r = 0 \tag{70}$$

in standard polarization (for the meaning of "standard polarization" here, see Section 5.1 in [4]). One can also think of these spectral curves in parametric form, as being given by the two following rational functions on $\mathbb{P}^1$:

$$x = \frac{z^r}{r}, \qquad y = -\frac{1}{z^{r-s}}. \tag{71}$$

Following through the steps of the correspondence established in Section 5 of [4], one sees that the value of the dilaton shift can be extracted as follows. One constructs the one-form

$$\omega_{0,1}(z) = y(z)dx(z) = -z^{s-1}dz. \tag{72}$$

The index $m$ of the mode $K_m$ that is shifted should be one more than the exponent of the power of $z$ in $\omega_{0,1}$, and the shift should be the coefficient. For instance, if one considered the quantum $r$-Airy structure of Theorem 4.9 but with dilaton shift

$$K_{-s} \mapsto K_{-s} - Q, \qquad Q \neq 0, \tag{73}$$

it would correspond to topological recursion on the spectral curve

$$x = \frac{z^r}{r}, \qquad y = -\frac{Q}{z^{r-s}}, \tag{74}$$

that is, on the algebraic curve

$$r^{r-s}x^{r-s}y^r - (-Q)^r = 0. \tag{75}$$

The coprime condition between $r$ and $s$ is crucial here: it ensures that the spectral curve, as an algebraic curve, is irreducible. The current formulation of topological recursion is only defined if the spectral curve is irreducible.

From the point of view of topological recursion, a natural question then is whether it is possible to generalize the definition of topological recursion to allow reducible algebraic spectral curves. We claim that the higher quantum Airy structures that we construct in this paper may give precisely such a generalization.

Let us be a little more precise. We consider the quantum $r$-Airy structures constructed in Theorem 3.5, originating from twisted modules of the underlying Heisenberg algebra with automorphisms given by $\sigma = \prod_{j=1}^n \sigma_j$, with each $\sigma_j$ a cycle of length $\rho$, and $n \geq 2$, and $r = n\rho$. We use the dilaton shifts by $r$-th roots of unity explored in Corollary 3.8:

$$K_{-s}^j \mapsto K_{-s}^j - \omega^j, \qquad j = 1, \ldots, n, \tag{76}$$

where $\omega$ is a primitive $r$-th root of unity. According to Theorem 3.5, we have two choices: either $s = 1$, or $s = 2$, $n = 2$, and $\rho$ is odd.

We claim that the case $s = 1$ should give an explicit formulation of topological recursion on the following reducible algebraic spectral curve:

$$\rho^{r-n}x^{r-n}y^r - (-1)^r = 0. \tag{77}$$

We note that this spectral curve is certainly reducible, as $r = n\rho$, and thus $r$ and $n$ are never coprime (since $n \geq 2$).

In fact, substituting $r = n\rho$, we can rewrite this curve in reduced form as:

$$\rho^{r-n}x^{r-n}y^r - (-1)^r = \prod_{j=1}^n \left(\rho^{\rho-1}x^{\rho-1}y^\rho - (-\omega^j)^\rho\right) = 0, \tag{78}$$

where $\omega$ is a primitive $r$-th root of unity. It has precisely $n$ components. It is now clear why we expect our quantum $r$-Airy structures to be connected to this spectral curve. Each component is a $(\rho, 1)$-spectral curve, but with dilaton shift $Q = \omega^j$. This is precisely what our construction is doing, with each cycle $\sigma_j$ of the automorphism $\sigma$ corresponding to an irreducible component of the reducible spectral curve.

More precisely, as topological recursion is not currently defined for reducible spectral curves, our claim is that:

*The quantum $r$-Airy structures of Theorem 3.5, with $\rho \geq 1$, $n \geq 2$, $s = 1$, $r = \rho n$, and the dilaton shifts being given by $r$-th roots of unity as in Corollary 3.8, may provide a definition of topological recursion on the reducible $(r, n)$-spectral curves (78). (Here, for the reducible $(r, n)$-spectral curve, $n \geq 2$ and $n \mid r$.)*

What about the other class of Theorem 3.5, with $n = 2$, $s = 2$, and $\rho$ odd? Following the same logic, it should define topological recursion on the reducible algebraic curves (here $r = 2\rho$):

$$\rho^{r-4} x^{r-4} y^r - (-1)^r = \prod_{j=1}^{2} \left( \rho^{\rho-2} x^{\rho-2} y^\rho - (-\omega^j)^\rho \right) = 0 \,, \tag{79}$$

where $\omega$ is a primitive $2\rho$-th root of unity. In other words, we claim that:

*The quantum $r$-Airy structures of Theorem 3.5, with $\rho \geq 1$, $\rho$ odd, $n = 2$, $s = 2$, $r = 2\rho$, and the dilaton shifts being given by $r$-th roots of unity as in Corollary 3.8, may provide a definition of topological recursion on the reducible $(r, 4)$-spectral curves (79). (Here, for the reducible $(r, 4)$-spectral curve, $r$ is even but $4 \nmid r$.)*

**Remark 3.10.** What is surprising however is that, with this construction, we do not recover all reducible $(r, n)$-spectral curves, but only these two particular families. It is unclear to us what is special about these families of reducible spectral curves, and why other families do not appear to have counterparts in our construction of quantum $r$-Airy structures.

**Example 3.11.** As an example, according to our claim, the quantum 4-Airy structure studied in Example 3.9 should correspond to the reducible $(4, 2)$-spectral curve:

$$4y^4 x^2 - 1 = \prod_{j=1}^{2} \left( 2y^2 x - (-i^j)^2 \right) = 0 \,. \tag{80}$$

# 4 Appending 1-cycles

In Section 3, we provided a classification of higher quantum Airy structures that arise as modules of $\mathcal{W}(\mathfrak{gl}_r)$-algebras following the method of [4], for arbitrary automorphisms $\sigma$ that are products of $n$ disjoint cycles of the same length. One can think of this construction as a natural generalization of Theorem 4.9 of [4], which considers the case $n = 1$ (i.e. $\sigma$ is the automorphism induced by the Coxeter element of the Weyl group).

Theorem 4.9 was also generalized in a different direction in [4]. Theorem 4.16 studied higher quantum Airy structures that can be obtained from automorphisms $\sigma$ that permute all but one of the basis vectors of the Cartan subalgebra. Moreover, in this context the extra one-cycle did not come with an extra dilaton shift. Thus, one may think of this result as follows. Given a quantum $r$-Airy structure constructed as a module of the $\mathcal{W}(\mathfrak{gl}_r)$-algebra as in Theorem 4.9, one can always construct a new quantum $(r + 1)$-Airy structure as a module of the $\mathcal{W}(\mathfrak{gl}_{r+1})$-algebra by "appending" to it a one-cycle, with no extra dilaton shift. This is, in essence, what Theorem 4.16 is doing.

In this section we investigate the question of whether, given a quantum $r$-Airy structure constructed as $\mathcal{W}(\mathfrak{gl}_r)$-module for an arbitrary automorphism $\sigma$, we can always construct a new quantum $(r+1)$-Airy structure as a $\mathcal{W}(\mathfrak{gl}_{r+1})$-module by appending to $\sigma$ a one-cycle, with no extra dilaton shift.

For $r \in \mathbb{Z}^+$, let $\Lambda_r$ be a $\lambda$-good index set for some integer partition $\lambda$ of $r$ (see Definition 2.2). Let $H_m^i$, $(i,m) \in \Lambda_r$, be the operators of a quantum $r$-Airy structure obtained as a $\mathcal{W}(\mathfrak{gl}_r)$-module descending from a twisted module of the underlying Heisenberg algebra, with the twist given by an automorphism $\sigma = \prod_{j=1}^n \sigma_j$, where the $\sigma_j$ are disjoint cycles of length $\rho_j$ respectively (and thus $r = \sum_{j=1}^n \rho_j$), and dilaton shifts

$$K_{-s_j}^j \mapsto K_{-s_j}^j - Q_j, \qquad j = 1, \ldots, n, \tag{81}$$

for some positive integers $s_j$ coprime with $\rho_j$. Suppose that $Q_j \neq 0$ for all $j \in \{1, \ldots, n\}$.

Now consider operators $\tilde{H}_m^i$ obtained as a $\mathcal{W}(\mathfrak{gl}_{r+1})$-module descending from a twisted module of the underlying Heisenberg algebra, with the twist $\tilde{\sigma}$ given by the same automorphism $\sigma$ but with an extra one-cycle appended, and the dilaton shifts still given by (81) (i.e. the extra bosonic modes associated to the one-cycle are not dilaton shifted). More precisely, if the $K_m^{r+1}$ are the bosonic modes associated to the extra one-cycle in $\tilde{\sigma}$, then the operators $\tilde{H}_m^i$ are obtained from the $H_m^i$ as follows:

$$
\begin{aligned}
\tilde{H}_m^1 &= K_m^{r+1} + H_m^1, \\
r^{i-1} \tilde{H}_m^i &= H_m^i + r \sum_{\substack{m_1, m_2 \in \mathbb{Z} \\ m_1 + m_2 = m-1}} K_{m_1}^{r+1} H_{m_2}^{i-1}, \qquad i = 2, \ldots, r, \\
r^r \tilde{H}_m^{r+1} &= r \sum_{\substack{m_1, m_2 \in \mathbb{Z} \\ m_1 + m_2 = m-1}} K_{m_1}^{r+1} H_{m_2}^r.
\end{aligned}
\tag{82}
$$

We set the extra bosonic zero-mode to zero: $K_0^{r+1} = 0$, and assume that $H_0^1 = 0$.

**Theorem 4.1.** *Suppose that there exists an integer partition $\tilde{\lambda}$ of $r+1$ such that $\tilde{\lambda}(i) = \lambda(i)$ for all $i = 1, \ldots, r$ and $\tilde{\lambda}(r+1) = \sum_{j=1}^n s_j$. Denote by $\tilde{\Lambda}_{r+1}$ the associated $\tilde{\lambda}$-good index set. Then the subset of operators $\tilde{H}_m^i$ constructed above, with $(i,m) \in \tilde{\Lambda}_{r+1}$, forms a quantum $(r+1)$-Airy structure.*

*Proof.* We need to determine whether conditions (1) and (2) of Definition 2.1 are satisfied for the set of operators $\tilde{H}_m^i$ with $(i,m) \in \tilde{\Lambda}_{r+1}$. Condition (2) is obviously satisfied by construction, since the set of modes $\tilde{H}_m^i$ is $\tilde{\lambda}$-good with respect to some partition $\tilde{\lambda}$ of $r+1$. What we need to check is whether condition (1) is satisfied. That is, we need to make sure that the $\tilde{H}_m^i$ have no degree zero terms, that the degree one terms only involve positive bosonic modes, and that there exist linear combinations of the modes such that all positive bosonic modes appear exactly once in the degree one terms.

Start with degree zero terms. Clearly, the operators $\tilde{H}_m^{r+1}$ cannot have degree zero terms. Now consider the operators $\tilde{H}_m^i$ with $i \in \{1, \ldots, r\}$. Since $\tilde{\lambda}(i) = \lambda(i)$ for all $i = 1, \ldots, r$, it means that for all $i = 1, \ldots, r$, $(i,m) \in \tilde{\Lambda}_{r+1}$ if and only if $(i,m) \in \Lambda_r$. In other words, we keep the same subset of modes for the $\tilde{H}_m^i$ as we did for the $H_m^i$, for $i = 1, \ldots, r$. As a result, since the $H_m^i$ with $(i,m) \in \Lambda_r$ have no degree zero terms (they form a quantum Airy structure), the $\tilde{H}_m^i$ with $(i,m) \in \tilde{\Lambda}_r$ for $i = 1, \ldots, r$ also have no degree zero terms.

We move on to degree one terms. We show first that they only involve positive bosonic modes. We start with the modes $\tilde{H}_m^1$, $m \geq 1$. We know that the modes $H_m^1$, $m \geq 1$ are in the original quantum Airy structure, and thus involve only positive bosonic modes. As a result,

the same is true for the modes $\tilde{H}_m^1$, $m \geq 1$. As for the zero mode, since $H_0^1$ is in the original quantum Airy structure, we must have $H_0^1 = 0$. We impose that $K_0^{r+1} = 0$, and therefore $\tilde{H}_0^1 = 0$.

Moving on to the $\tilde{H}_m^i$ with $i \in \{2, \ldots, r\}$, inspecting (82) we see that there are two sources of degree one terms: the degree one terms from the $H_m^i$, and potential degree one terms arising from degree zero terms of $H_{m_2}^{i-1}$ in

$$\sum_{\substack{m_1, m_2 \in \mathbb{Z} \\ m_1 + m_2 = m-1}} K_{m_1}^{r+1} H_{m_2}^{i-1} \,. \tag{83}$$

Since the $H_m^i$ are in a quantum Airy structure, we know that their degree one terms only involve positive bosonic modes. As for the second source of degree one terms, they will only involve positive bosonic modes $K_{m_1}^{r+1}$, $m_1 > 0$, if the $H_{m_2}^{i-1}$ with $m_2 \geq m$ do not have degree zero terms. (Here we use the fact that $K_0^{r+1} = 0$.) But the $H_{m_2}^{i-1}$ with $m_2 \geq m$ are part of the original quantum Airy structure. Indeed, if $(i, m) \in \Lambda_r$, then $m \geq i - \lambda(i)$. But for any partition, either $\lambda(i) = \lambda(i-1)$ or $\lambda(i) = \lambda(i-1) + 1$. Either way, $\lambda(i) \leq \lambda(i-1) + 1$, and thus $m \geq i - \lambda(i) \geq i - 1 - \lambda(i-1)$, which means that $(i-1, m) \in \Lambda_r$. Thus the $H_{m_2}^{i-1}$ with $m_2 \geq m$ have no degree zero terms. We conclude that the degree one terms of the $\tilde{H}_m^i$ with $i \in \{2, \ldots, r\}$ and $(i, m) \in \tilde{\Lambda}_{r+1}$ involve only positive bosonic modes.

Consider finally the modes $\tilde{H}_m^{r+1}$. The only potential degree one terms arise from degree zero terms of $H_{m_2}^r$ in

$$\sum_{\substack{m_1, m_2 \in \mathbb{Z} \\ m_1 + m_2 = m-1}} K_{m_1}^{r+1} H_{m_2}^r \,. \tag{84}$$

Using the same argument as above, it is clear that those potential degree one terms will also only involve positive bosonic modes.

Finally, we need to make sure that there exist linear combinations of the modes such that all positive bosonic modes appear exactly once in the degree one terms. Consider first the degree one terms in the modes $\tilde{H}_m^{r+1}$. It is fairly straightforward to calculate that

$$H_{m_2}^r = A \left( \prod_{j=1}^n (-Q_j)^{\rho_j} \right) \delta_{m_2, r - \sum_j s_j - 1} + \mathcal{O}(1) \,, \tag{85}$$

for some number $A$. As a result, $\tilde{H}_m^{r+1}$ has a degree one term of the form

$$r^r \tilde{H}_m^{r+1} = rA \left( \prod_{j=1}^n (-Q_j)^{\rho_j} \right) K_{m-r+\sum_j s_j}^{r+1} + \mathcal{O}(2) \,. \tag{86}$$

Since we assume that all $Q_j \neq 0$, this term is non-vanishing. Moreover, since the partition $\tilde{\lambda}$ is such that $\tilde{\lambda}(r+1) = \sum_{j=1}^n s_j$, we know that the $\tilde{\lambda}$-good subset of modes only contain the modes $\tilde{H}_m^{r+1}$ with $m \geq r + 1 - \tilde{\lambda}(r+1) = r + 1 - \sum_{j=1}^n s_j$. We conclude that all positive modes $K_k^{r+1}$ with $k \geq 1$ appear exactly once in the degree one terms of the $\tilde{H}_m^{r+1}$ in the $\tilde{\lambda}$-good subset.

For the remaining modes $\tilde{H}_m^i$ with $i \in \{1, \ldots, r\}$, the degree one terms always consist of the degree one term of $H_m^i$ plus some linear combination of positive modes $K_k^{r+1}$, $k \geq 1$. As the $H_m^i$ form a quantum Airy structure, we know that all positive bosonic modes associated to the cycles of $\sigma$ appear exactly once in the linear terms of the $H_m^i$, and hence of the $\tilde{H}_m^i$. Then, by taking linear combinations with the $\tilde{H}_m^{r+1}$, we can remove the modes $K_k^{r+1}$ from the linear terms. We conclude that condition (1) of Definition 2.1 is satisfied, and that the $\tilde{H}_m^i$ form a quantum Airy structure. $\qquad \square$

**Remark 4.2.** In the construction above we set the extra bosonic zero-mode to zero $K_0^{r+1} = 0$, and assumed that $H_0^1 = 0$ so that $\tilde{H}_0^1 = 0$. However, it may be interesting to generalize the construction by letting $K_0^{r+1} = \hbar^{1/2}q = -H_0^1$ for some $q \in \mathbb{C}$, such that $\tilde{H}_0^1 = 0$. This is for instance what was considered in the special case of Theorem 4.17 in [4]. Looking back at the proof of the Theorem above, in particular the paragraph below (83), we note that this will be allowed if the modes $H_{i-\tilde{\lambda}(i)-1}^{i-1}$, for $i \in \{2, \ldots, r+1\}$, do not have degree zero terms. This is necessary so that $K_0^{r+1}$ does not appear as a degree one term in the $\tilde{H}_m^i$. However, the modes $H_{i-\tilde{\lambda}(i)-1}^{i-1}$ may or may not be in the original quantum Airy structure (they are if $\tilde{\lambda}(i) = \tilde{\lambda}(i-1)$, but they are not if $\tilde{\lambda}(i) = \tilde{\lambda}(i-1) + 1$). If they are, then they certainly do not have degree zero terms, but if they are not, then it is unclear a priori whether they have degree zero terms. Nevertheless, if for a given example one can check that these modes do not have degree zero terms, then the construction above can be extended by letting $K_0^{r+1} = \hbar^{1/2}q = -H_0^1$ for some $q \in \mathbb{C}$.

As an example, we can apply this theorem to the higher quantum Airy structures constructed in Theorem 3.5.

**Corollary 4.3.** *Consider the quantum $r$-Airy structures constructed in Theorem 3.5. Assuming that all dilaton shifts $Q_j$, $j = 1, \ldots, n$ are non-zero, a one-cycle can be appended to the quantum $r$-Airy structures as in Theorem 4.1 in cases (a) and (b) to produce new quantum $(r+1)$-Airy structures, but not in cases (c) and (d).*

*Proof.* This follows by inspection of the partitions.

Case (a) consists of an automorphism $\sigma$ which consists of $r$ disjoint one-cycles, with $s_j = 1$ for all $j = 1, \ldots, r$. The partition of $r$ is $\lambda = (2, 1, \ldots, 1)$. Then we must have $\tilde{\lambda} = (2, 1, \ldots, 1, 1)$, which is a partition of $r+1$ such that $\tilde{\lambda}(i) = \lambda(i)$ for all $i = 1, \ldots, r$ and $\tilde{\lambda}(r+1) = r = \sum_{j=1}^r s_j$.

Case (b) consists of an automorphism $\sigma$ which consists of $n$ disjoint $\rho$-cycles, with $s_j = 1$ for all $j = 1, \ldots, n$. The partition of $r = n\rho$ is $\lambda = (\rho+1, \rho, \ldots, \rho, \rho-1)$. Then we can construct a partition $\tilde{\lambda}$ of $r+1$ as $\tilde{\lambda} = (\rho+1, \rho, \ldots, \rho, \rho)$. It is such that $\tilde{\lambda}(i) = \lambda(i)$ for all $i = 1, \ldots, r$, and $\tilde{\lambda}(r+1) = n = \sum_{j=1}^n s_j$.

However, this doesn't work for cases (c) and (d). Case (c) corresponds to an automorphism $\sigma$ which consists of two disjoint one-cycles, with $s_j = 2$ for both. The partition of $r = 2$ is $\lambda = (1, 1)$. The only possibility for the partition $\tilde{\lambda}$ of $r+1 = 3$ such that $\tilde{\lambda}(i) = \lambda(i)$ for $i = 1, 2$ is $\tilde{\lambda} = (1, 1, 1)$, but then $\tilde{\lambda}(3) = 3$ which is not equal to $\sum_{j=1}^2 s_j = 4$.

Case (d) corresponds to two disjoint $\rho$-cycles, with $\rho$ odd, and $s_j = 2$ for both cycles. The partition $\lambda$ of $r = 2\rho$ is $\lambda = \left(\frac{\rho+1}{2}, \frac{\rho+1}{2}, \frac{\rho-1}{2}, \frac{\rho-1}{2}\right)$. The only choice for the partition $\tilde{\lambda}$ of $r+1 = 2\rho+1$ such that $\tilde{\lambda}(i) = \lambda(i)$ for $i = 1, \ldots, r$ is $\tilde{\lambda} = \left(\frac{\rho+1}{2}, \frac{\rho+1}{2}, \frac{\rho-1}{2}, \frac{\rho-1}{2}, 1\right)$, but then $\tilde{\lambda}(r+1) = 5$, which is not equal to $\sum_{j=1}^2 s_j = 4$. $\square$

# 5 Future directions

In this paper we made a first step towards a classification of higher quantum Airy structures constructed as $\mathcal{W}(\mathfrak{gl}_r)$-modules following the method of [4], by classifying those that arise from twisted modules of the underlying Heisenberg algebra with the twist corresponding to an automorphism with arbitrary disjoint cycles of the same length. We also studied the question of when new higher quantum Airy structures can be constructed by "appending a one-cycle with no dilaton shift".

A few open questions immediately come to mind:

- It would be very interesting to complete the classification for arbitrary automorphisms $\sigma$. The key insight that the degree condition (2) can be thought of as a matrix inversion problem could prove useful.

- The proposed interpretation of our quantum $r$-Airy structures as defining topological recursion on reducible spectral curves in Section 3.6 deserves to be studied further. For instance, a residue formulation of topological recursion on reducible spectral curves could potentially be extracted from our quantum $r$-Airy structures. The fact that only particular families of reducible spectral curves seem to have counterparts in our classification of quantum $r$-Airy structures is also intriguing and deserves further investigation.

- The quantum $r$-Airy structures constructed as $\mathcal{W}(\mathfrak{gl}_r)$-modules for fully cyclic automorphisms, as in Theorem 4.9 of [4], have natural interpretations in enumerative geometry. They are known to produce generating functions for various flavours of (closed) intersection theory on $\overline{\mathcal{M}}_{g,n}$ (or variants thereof). It would be interesting to explore whether the quantum $r$-Airy structures for more general automorphisms, such as those constructed in Theorem 3.5, also have interesting enumerative geometric interpretations.

- The idea of "appending a one-cycle with no dilaton shift" to a higher quantum Airy structure has a compelling interpretation in enumerative geometry, for the particular families of higher quantum Airy structures studied in [4]. Indeed, while the higher quantum Airy structures for fully cyclic automorphisms from Theorem 4.9 are connected to various flavours of closed intersection theory on $\overline{\mathcal{M}}_{g,n}$ (or variants thereof), the corresponding higher quantum Airy structures obtained by appending a one-cycle, as in Theorem 4.16, are related to the open version of the appropriate intersection theory. "Appending a one-cycle" to the higher quantum Airy structures may then be understood as some sort of open/closed correspondence. If an enumerative geometric interpretation for higher quantum Airy structures for arbitrary automorphisms is found, it would be fascinating to see whether such an open/closed correspondence holds for the general procedure of appending a one-cycle studied in Theorem 4.1.

## Acknowledgments

We thank Gaetan Borot, Nitin K. Chidambaram, and Thomas Creutzig for discussions, as well as Devon Stockall and Quinten Weller for collaboration in the initial stages of this project.

**Funding information** The authors acknowledge the support of the Natural Sciences and Engineering Research Council of Canada (NSERC); in particular, the work of K.M. was partly supported by a NSERC Undergraduate Student Research Award.

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
