# Peer review of "A New Class of Higher Quantum Airy Structures as Modules of $\mathcal{W}(\mathfrak{gl}_r)$-Algebras"

_SciPost Physics, doi:SciPost Phys. 14, 169 (2023)_

## Round 1 · Referee Report · Anonymous (Referee 1) · 2022-4-3

Report

This paper is a follow-up of a previous paper co-authored by one of the authors, in which the concept of higher quantum Airy structures was introduced. This quadratic quantum Airy structures introduced by Kontsevich and Soibelman generalize the Virasoro constraints on the tau-functions related to matrix models and to the theory of intersections, while the higher quantum Airy structures generalize the W-constraints. In the previous paper, the quantum Airy structure was associated with a W-algebra for the Lie-algebra gl(n) and an automorphism induced by the Coxeter element of the Weyl group. Such quantum Airy structure describes the critical points of the two-matrix model. In the present paper, a more general automorphism with several cycles of the same length is considered. A classification of the higher quantum Airy structures obtained in this way is presented and some examples are given. The relation to the topological recursion is discussed, and a conjecture is advanced that the generalisation considered corresponds to topological recursion associated with a reducible algebraic curve. An interesting point not discussed in the paper is which type large-N matrix integrals are described by reducible algebraic curves.
I think that this paper contains original results and contributes to a larger program of finding a complete classification of the higher quantum Airy structures and their relation to the topological recursion. I recommend publication.

---

## Round 1 · Referee Report · Anonymous (Referee 2) · 2023-5-13

# A NEW CLASS OF HIGHER QUANTUM AIRY STRUCTURES AS MODULES OF $\mathcal{W}(\mathfrak{gl}_r)$-ALGEBRAS

May 12, 2023

## Summary

This article is an extension of ideas from its reference [4] by Borot, Bouchard, Chidambaram, Creutzig and Noshchenko. Its origin is Kontsevich and Soibelman's introducing of quantum Airy structures. They generalize the topological recursion, in the sense that instances of the topological recursion give rise to quantum Airy structures. In ref [4], the authors first extended the quantum Airy structures to higher quantum Airy structures and showed that they generalize the generalized topological recursion of Bouchard-Eynard for spectral curves with non-simple branch points. They moreover constructed examples of higher quantum Airy structures using W-algebras.

The present article is a continuation of this latter idea. In [4], $\mathcal{W}(\mathfrak{gl}_r)$-modules leading to higher quantum Airy structures were built using specific "twists", which are automorphisms of the Weyl group of $\mathfrak{gl}_r$. Their twists were a single cycle. Here the authors consider different twists where the automorphisms are products of cycles of the same length. The main theorem is Thm 3.5 which shows that new higher quantum Airy structures can be obtained with those twists, and they are classified in four types and depend on a "shift matrix" and some zero mode constants.

It is further argued that they should generalize the topological recursion on reducible spectral curves. However the TR has never been defined on reducible spectral curves, so the present results could in fact be used to do so. A final section is devoted to appending a 1-cycle to the automorphism, thus going from $r$ to $r+1$. In [4], this was done and argued as going from closed to open intersection theory, which is a motivation to do so here as well as, although this interpretation is not known in the present cases.

## Recommendation

The classification of quantum Airy structures is an important goal for enumerative geometry. While the main ideas of this article are from [4], it is an interesting continuation of it and answers natural questions since [4] left open many cases for the automorphism $\sigma$. In some sense, it also successfully tests the methods of [4]. The conjectured relation to reducible spectral curves is quite convincing. Practical examples are produced by choosing an explicit shift matrix in Section 3.5, with a fully explicit example at $r = 4$.

The article is well written. Although prior knowledge of [4] is highly preferred, I did not find that it was necessary to know [4] in full details to follow this article. The method is well explained and the steps are mostly easy to follow. I therefore strongly recommend it for publication in Scipost.

## Corrections

I have some minor suggestions and some needed clarification, which do not jeopardize the publication and should be easily addressed by the authors.

1. I guess that the $\sigma_j$ have disjoint support on $\{1, \ldots, r\}$ but I did not find it explicitly written.

2. Def. 2.1 Why the subscript W for the "completed algebra of differential operators on $V$"? Is it $V \oplus V^*$ like in [4]?

3. In definition 2.2 please do not use additions for integer partitions, this is way too confusing (for example, in the definition of $\lambda(i)$ there is a sum of parts, which is here not a partition!). Use instead a standard convention like (see Macdonald's book on symmetric functions for example, or just reference [4]): $\lambda = (\lambda_1, \lambda_2, \dots)$ with $\lambda_1 \geq \lambda_2 \geq \cdots \geq 0$ is a partition of $r$, i.e. $\sum_i \lambda_i = r$. Also it is not said here whether $\lambda_1 \geq \lambda_2 \geq \cdots \geq 0$.

   A few equations should be corrected like (3.25) which becomes $\lambda = (2, 1^{n-2})$, (3.30) $\lambda = (\rho + 1, \rho^{n-2}, \rho - 1)$, (3.34) $\lambda = (1, 1)$, (3.37), also in the proof of Corollary 4.2.

4. Inverse missing in eq. (2.5)

5. What is the range of the indices of $H_m^i$ in general?

6. The authors keep writing "pick a subalgebra of the modes $W_m^i$" for instance in step 2 below Def. 2.2. It is more precise in Corollary 3.2 with the constraint (3.12). It is also apparent in Lemma 3.4 and its proof and so on. What does subalgebra mean in that context and why (3.12) still produces a subalgebra? The property of being $\lambda$-good is not checked as those points but maybe it that nonetheless?

7. typo in "there exists linear combinations", below (3.13) and also before (3.17), and in Lemma 3.14, etc.

8. What is $d$, below (3.17)? Is it $n$, because $l \in \{0, ..., n-1\}$ has $n$ possible values?

9. Typo in "there is a much larger choices of $s$"

10. Below (3.33) you refer to Section 3.3. in [4]. Can you be more precise?

11. I am not sure I understand (3.39). In a partition, the parts are stored in decreasing order just so that there is a unique way to writing it. But if there is an $l$ such that $\lambda_{l+1} > \lambda_l$, one typically switch them. Is it not possible here? Maybe explaining how the partition is found could be helpful. Are the parts found inductively? In any case, please clarify (3.39). I guess this is the same for (3.45).

12. Below (3.72), it is said to focus on $\rho > 1$. I would remove the parentheses around the next sentence on $\rho = 1$. The case $\rho = 1$ was treated the same as $\rho > 1$ in Theorem 3.5, so mentioning that it however does not seem to be relevant for the topological recursion is important enough to deserve more than parentheses. Is it also possible to expand a bit on why?

13. For the paragraph below (4.2) on the degree condition for $\tilde{H}_m^i$, I do not understand where terms of degree 0 could be in (4.2), and what it means that the subalgebra of tilde modes may be smaller than that of $H_m^i$.

14. Same with eq. (4.3), I do not understand those terms of degree 0 if $H_{m_2}^r$ come from a higher Airy structure.

---

## Round 2 · Author Response

The authors thank the referees for constructive comments and suggestions.

---

## Round 2 · List of Changes

- Minor revisions throughout the paper.
- Remark (with reference) added in the introduction.
- Minor rewriting of the proofs of Theorems 3.5 and 4.1.
- Remark 4.2 added.

You are currently on this page

Resubmission 2009.13047v2 on 31 May 2023

---

## Editorial Decision

published